# Investigation on ABCC6-Deficient Human Hepatocytes Generated by CRISPR–Cas9 Genome Editing

**DOI:** 10.3390/cells14080576

**Published:** 2025-04-11

**Authors:** Ricarda Plümers, Svenja Jelinek, Christopher Lindenkamp, Michel R. Osterhage, Cornelius Knabbe, Doris Hendig

**Affiliations:** Herz- und Diabeteszentrum Nordrhein-Westfalen, Institut für Laboratoriums- und Transfusionsmedizin, Universitätsklinik der Ruhr-Universität Bochum, Medizinische Fakultät OWL (Universität Bielefeld), Georgstraße 11, 32545 Bad Oeynhausen, Germany

**Keywords:** ABCC6, pseudoxanthoma elasticum, hepatocytes, CRISPR Cas

## Abstract

Patients affected by the rare disease pseudoxanthoma elasticum (PXE) exhibit the calcification of elastic fibers in ocular, dermal, and vascular tissues. These symptoms are triggered by mutations in the ATP-binding cassette transporter subfamily C member 6 (ABCC6), whose substrate remains unknown. Interestingly, ABCC6 is predominantly expressed in the liver tissue, leading to the hypothesis that PXE is a metabolic disorder. We developed a genome-editing system targeting ABCC6 in human immortalized hepatocytes (HepIms) for further investigations. The HepIms were transfected with an *ABCC6*-specific clustered regulatory interspaced short palindromic repeat (CRISPR-Cas9) genome-editing plasmid, resulting in the identification of a heterozygous (*ht^ABCC6^*HepIm) and a compound heterozygous (*cht^ABCC6^*HepIm) clone. These clones were analyzed for key markers associated with the PXE pathobiochemistry. Hints of impaired lipid trafficking, defects in the extracellular matrix remodeling, the induction of calcification inhibitor expression, and the down regulation of senescence and inflammatory markers in *ABCC6*-deficienct HepIms were found. Our *ABCC6* knock-out model of HepIms provides a valuable tool for studying the metabolic characteristics of PXE in vitro. The initial analysis of the clones mirrors various features of the PXE pathobiochemistry and provides an outlook on future research approaches.

## 1. Introduction

The symptoms of the progressive, rare genetic disease pseudoxanthoma elasticum (PXE, OMIM #264800) appear in the skin, the vascular walls, and the retina (Bruch’s membrane) and are caused by ectopic calcification in the elastic fibers [1]. This results in a loss of the elasticity of the skin, progressive loss of vision, and atherosclerotic changes in the vessels of patients [2,3,4]. Surprisingly, mutations in the gene of ATP-binding cassette transporter subfamily C member 6 (ABCC6), a transporter with a predominant expression in the liver, have been identified as the cause of PXE [5]. The metabolic hypothesis was developed according to the discrepancy between the ABCC6 expression and the localization of symptoms in terms of systemic distribution. It claims that the deficiency of the unknown substrate in the bloodstream, due to restricted transport from the liver by ABCC6, evokes the clinical manifestation [6].

Ectopic calcification is the final mediator of the PXE symptoms. The dysregulation of the limited physiological calcification results in the formation of hydroxyapatite crystals [7]. The local formation of these crystals is largely determined by the presence of the anti-calcifying inorganic pyrophosphate (PP_i_) [8]. The concentration of PP_i_ depends on the conversion of ATP to AMP and PP_i_ by ectonucleotide phosphodiesterase (ENPP) [9]. In addition, systemic expressed factors, such as matrix gla protein (MGP), circulating in the blood also have an inhibitory effect on crystal growth [10].

Elastic fibers, which appear calcified and fragmented in PXE patients’ skin biopsies, are composed of a core of polymerized elastin covered by fibrillin (FBN) [11]. Other macromolecules are also affected by structural changes upon *ABCC6* deficiency and are subject to an extensive degradation by matrix metalloproteinases (MMPs) [12,13]. These macromolecules include collagen fibrils [14]. Their assembly is supported by another group of extracellular matrix (ECM) components, the proteoglycans, to which decorin (DCN) belongs [15].

Metabolic analyses of *ABCC6*-deficient fibroblasts and adipocytes revealed alterations in the lipid metabolism of cholesterol- and triglyceride (TG)-containing lipoproteins [16,17]. Exemplarily, a reduced low-density lipoprotein receptor (LDLR) expression has been observed. The binding of low-density lipoprotein (LDL) to its receptor is inhibited by proprotein convertase subtilisin/kexin type 9 (PCSK9) [18]. The uptake of other lipids occurs via the fatty acid transporter cluster of differentiation 36 (CD36) or the high-density lipoprotein receptor scavenger receptor class B member 1 (SCARB1) [19,20]. Cholesterol absorbed into the liver can be further metabolized by enzymes, such as sterol 27-hydroxylase (CYP27A1), and secreted as bile acid [21]. Furthermore, *Abcc6* deficiency in the mouse model affected the expression pattern of other lipids and bile acid transferring ABC transporters [22]. These included ABCA1, a transporter in the reverse transport of cholesterol from the liver to the periphery, as well as ABCC2 and ABCC3, which secrete bile acids from the liver into the bile ducts [23].

Markers of cellular senescence and inflammation have recently been detected in plasma samples obtained from PXE patients and in their fibroblasts. The early senescence is thought to be dependent on the cell cycle inhibitor p21 (CDKN1A), leading to the formation of a senescence-associated secretory phenotype [24]. This secretome includes the proinflammatory cytokines interleukin 6 (IL6) and interleukin 1β (IL1B) [25,26]. Furthermore, the overexpression of the complement system and its components, C1r and C3, was detected in PXE fibroblasts [27].

Our aim was to develop an *ABCC6*-deficient hepatocyte cell culture system. The use of immortalized hepatocytes (HepIms), i.e., primary hepatocytes whose aging is prevented by the insertion of specific genes, has been established as a model that largely maintains the characteristics of primary hepatocytes while being long living enough for gene-modifying experimental approaches [28,29,30,31]. Using a clustered, regularly interspaced short palindromic repeats (CRISPR) Cas9-based genome-editing system, we generated *ABCC6*-deficient HepIm clones and performed a preliminary analysis of their characteristics regarding the PXE pathobiochemistry.

## 2. Materials and Methods

### 2.1. Human Hepatocyte Cell Culture

Commercially available immortalized human hepatocytes (HepIm; catalog number T0063) were obtained from abm good (Richmond, BC, Canada). The cells were cultured according to the supplier’s instructions in standard cell culture dishes with Prigrow IX medium (abm good, Richmond, BC, Canada) supplemented with 1% (*v*/*v*) penicillin–streptomycin–amphotericin B solution (100×; PAN Biotech, Aidenbach, Germany) and 10% (*v*/*v*) fetal calf serum (FCS; gibco, Waltham, MA, USA). Cultivation was carried out at 37 °C incubation with 5% CO_2_. The medium was changed every two to three days and hepatocytes were subcultured upon confluence, utilizing 0.05% (*v*/*v*) trypsin (PAN Biotech, Aidenbach, Germany).

In preparation of experiments, the hepatocytes were seeded at a density of 500 cells per mm^2^. Upon 24 h cultivation in complete medium, the nutrient liquid was changed to Prigrow IX medium containing 10% (*v*/*v*) delipidated FCS (gibco, Waltham, MA, USA) followed by an assay-dependent incubation period of 72 h or 21 d. The delipidation of FCS was achieved following the protocol of Gibson et al. [32]. In brief, 50 mL of FCS and 1 g of silicic acid powder were mixed and incubated at 4 °C overnight. The silicic powder and bound lipid were separated from the delipidated FCS by centrifugation of 5000× *g* for 1 h at 4 °C. The delipidated FCS was sterile filtered (0.2 µm) subsequently.

### 2.2. CRISPR–Cas9 Mediated hABCC6 Knock-Out

A CRISPR–Cas9-based knock-out in HepIm was targeted using a plasmid-based system. Preliminary tests showed that transfection of the hepatocytes with a ribonuclein–protein complex, which worked well for other cell lines, was not possible [17]. The choice of transfection reagent was also based on a comparison of different reagents. An *hABCC6*-specific guide RNA (gRNA), displayed in Table 1, was cloned into the PX458 vector (Addgene, Watertown, MA, USA).

One day prior to transfection, HepIms were seeded under standard conditions at a density of 1.8 × 10^4^ cells per 6-well culture plate. Shortly before the transfection, the medium was switched for an antibiotic-free medium. An amount of 12 µL of FuGene 6 transfection reagent (Promega, Madison, WI, USA) was incubated with 188 µL Opti-MEM for 5 min at room temperature (RT) before adding 4 µg of gRNA containing vector PX458. The mixture was incubated for 30 min at RT and subsequently spotted onto the HepIm. The medium was changed 24 h later to antibiotic-containing culture medium. Another 24 h later, the cells were separated via fluorescence-associated cell sorting based on the expression of green fluorescent protein encoded by PX458. Sorting was performed on the S3e Cell Sorter (Bio-Rad, Hercules, CA, USA) by differentiation through the FITC channel (Fl1, λex/em = 495/519). Afterwards, cells were transferred into a 96-well culture plate at a density of 1 cell per well. A few hours later, the wells were checked for single attached cells. Single clone cultures were successively transferred into higher well formats and finally cryopreserved. Wildtype control HepIms were treated in the same manner, except for the addition of the *ABCC6*-specific gRNA containing vector. In general, the transfection efficiency as well as gene editing efficiency were low (<1%) due to the poor transfectability of hepatocytes. Therefore, the decision was made not to use a cell pool but available individual clones for the following analysis.

### 2.3. Nucleic Acid Isolation

Isolation of nucleic acids from cell cultures was performed with the NucleoSpin Blood Extraction Kit (Macherey-Nagel, Düren, Germany) for DNA isolation and the NucleoSpin RNA Kit (Macherey-Nagel, Düren, Germany) for RNA isolation, according to the manufacturer’s instructions. Concentrations of nucleic acid were determined with a NanoDrop 2000 spectrophotometer (Peqlab, Erlangen, Germany).

### 2.4. Genomic DNA Amplification

The amplification of a desired sequence was achieved with the HotStarTaq DNA polymerase kit (Qiagen, Hilden, Germany). Site-specific primer sequences can be found in Table 2. A volume of 3 µL of the DNA template was added to a mixture of 29.3 µL water, 5 µL PCR buffer, 10 µL Q-Solution, 0.5 µL dNTPs (10 mM each), 1 µL 25 µM forward and reverse primers (Biomers, Ulm, Germany), and 0.2 µL HotStar Taq polymerase. The program for PCR comprised an initial denaturation (15 min, 95 °C) followed by 35 cycles of denaturation (1 min, 95 °C), annealing (1 min, primer pair-specific annealing temperature T_A_), and elongation (1 min, 72 °C), ending with a final elongation (10 min, 72 °C).

Amplificates were purified with the MSB Spin PCRapace Kit (Invitek Molecular, Berlin, Germany), according to manufacturer’s instructions.

### 2.5. T7 Endonuclease/Mutagenesis Assay

T7 endonuclease I, which cleaves base pair mismatches, and its 10× reaction buffer were purchased from NEB (Ipswich, MA, USA). A quantity of 200 ng of genomic DNA amplificate filled up to 17 µL with water was mixed with 2 µL 10× reaction buffer, heated to 95 °C, and gradually cooled at a rate of −0.3 °C per s to 25 °C to form heteroduplexes. The mixture was further incubated for 1 h at 37 °C after the addition of 1 µL of T7 endonuclease-I (10 U/µL). The reaction was stopped subsequently with 1.5 µL of a 0.25 M EDTA solution. Resulting DNA fragments were visualized by electrophoresis through a 1.8% agarose gel with ethidiumbromide. The pUC19 DNA Mspl (Hpall) Marker (Thermo Fisher Scientific, Waltham, MA, USA) was used as a length standard.

### 2.6. TA Cloning for Single Allele Sequencing

The TA cloning assay from Thermo Fisher Scientific (Waltham, MA, USA) was used. Considering the poly-A-overhangs generated during DNA amplification, 1.5 µL of purified amplificates were ligated into 2.0 µL of linear pCR 2.1 vector (25 ng/µL) under the addition of 2 µL T4 ligase buffer, 3.5 µL water, and 1 µL ExpressLink T4 Ligase (5 U/µL) at 4 °C overnight. The plasmids were transformed into TOP10 Escherichia coli through the heat shock method (30 min on ice, 30 s at 42 °C, and 2 min on ice). Resuspended in 500 µL of S.O.C. medium (Thermo Fisher Scientifc, Waltham, MA, USA), the bacteria were incubated at 37 °C for 1 h. Afterwards, they were plated onto LB agar plates containing 75 µg/mL of ampicillin. Liquid bacterial cultures in 5 mL of LB medium with 75 µg/mL of ampicillin originating from single bacterial clones on LB plates were used for plasmid isolation with the QIAprep Spin Miniprep Kit (Qiagen, Hilden, Germany).

### 2.7. Sanger Sequencing

Sequencing reactions were prepared with the BigDye Terminator v3.1 Cycler Sequencing Kit (Thermo Fisher Scientific, Waltham, MA, USA). For each sequencing reaction, either 2.5 µL genomic DNA amplification product or 200–300 ng of plasmid were combined with 2 µL of premix, 1.5 µL 2.5 µM forward primer, and 4 µL BigDye buffer (5×) and filled up to a total volume of 20 µL with water. The reaction occurred at 95 °C for 2 min (initial denaturation) followed by 30 cycles of 95 °C for 10 s (denaturation), primer-specific annealing temperature for 10 s (annealing), and 60 °C for 4 min (elongation). The resulting mixtures were purified by centrifugation at 1000× *g* for 4 min through a Sephadex G-50 column (Sigma-Aldrich, St. Louis, MO, USA). Sanger sequencing was conducted with the ABI Prism 3500 Genetic Analyzer (Thermo Fisher Scientific, Waltham, MA, USA).

### 2.8. Gene Expression Analysis

Complementary DNA was transcribed from 1 µg of RNA with the SuperScript II Kit (Thermo Fisher Scientific, Waltham, MA, USA). qRT-PCR was performed using reaction mixtures containing 5 µL of LightCycler 480 SYBR Green I Master Reagent (Roche, Basel, Switzerland), 0.25 µL of 25 µM forward and reverse primers (Biomers, Ulm, Germany), 2 µL of water, and 2.5 µL of 1:10 diluted complementary DNA. Table 3 lists the primer sequences. The PCR was conducted on a LightCycler 480 Instrument II system (Roche, Basel, Switzerland). The program included an initial incubation for 5 min at 95 °C, 45 cycles of denaturation (95 °C, 10 s), annealing (specific annealing temperature, 15 s), and extension (72 °C, 20 s) followed by a melting curve analysis. The relative normalization of gene-of-interest expression was carried out based on the expression of the housekeeping genes relative to succinate dehydrogenase complex flavoprotein subunit A (SDHA), hydroxymethylbilane synthase (HMBS), and ribosomal protein L13 (RPL13). Calculations were based on the ΔΔC_t_ method, considering PCR efficiency and internal calibration. For each condition, biological and technical triplicates were prepared.

### 2.9. Immunoassay for Protein Detection

The PCSK9, C1r, and C3 protein levels in cell culture supernatants were analyzed via commercially available ELISA Kits, according to the manufacturer’s instructions (DPC900, R&D Systems, Abingdon, UK; ab170245 and ab108823, Abcam, Cambridge, UK). The IL6 levels in the cell culture supernatant were determined using the immunoanalyzer Cobas e411 (Roche, Basel, Switzerland). Normalization was performed based on the DNA content in the cell lysate.

### 2.10. Immunofluorescent Staining and Fluorescence Microscopy

The cells were cultured on cover slips in 24-well plates for immunocytochemical detection of proteins. After two washes with phosphate-buffered saline (PBS), the cells were fixed with 400 µL of 4% PFA for 20 min on ice. Subsequent incubation of the cells with 400 µL of 0.1% Triton X-100 in PBS permeabilized the cell membranes. The cells were washed with PBS two times. Afterward, they were incubated for 1 h at RT with 200 µL of a 5% bovine serum album (BSA) solution in PBS to block free binding sites. Following further washing steps, 50 µL of the primary antibody solution in a 1% BSA solution in PBS was applied followed by an incubation for 2 h at RT. After another washing step, the cells were incubated for one hour with 50 µL of the secondary antibody solution at a fourfold higher dilution compared to the primary antibody solution in 1% BSA in PBS. Two further washes were followed by an incubation with 200 µL of 0.25 µM DAPI solution in PBS. After two final washes, the coverslips were covered with ROTI Mount FluorCare mounting medium (Roth, Karlsruhe, Germany) and transferred to microscope slides. The preparations were then examined on a BZ-X810 fluorescence microscope (Keyence, Osaka, Japan). Analysis was performed using the BZ-810 analyzer (Keyence, Osaka, Japan) and ImageJ (version 1.54c, National Institutes of Health, Bethesda, MD, USA).

### 2.11. SA-β-Galactosidase Assay

The hydrolysis of 4-methylumbelliferyl-D-galactopyranoside (MUG) to 7-hydroxy-4-methylcoumarin by SA-β-galactosidase can be determined photometrically and correlates with the activity of the enzyme [33]. Cells were harvested with 300 µL of lysis buffer containing 0.2 M of disodium hydrogen phosphate, 0.1 M of citric acid, 5 mM of 3-[(3-cholamidopropyl)dimethylammonio]-1-propanesulfonate, 0.5 mM of benzamidine, and 0.25 mM of phenylmethylsulfonyl fluoride. The lysates were vortexed and centrifuged at 12,000× *g* and 4 °C for 5 min. An amount of 100 µL of the supernatant was combined with an equivalent portion of the reaction buffer which contained 0.2 M of disodium hydrogen phosphate, 0.1 M of citric acid, 300 mM of sodium chloride, 10 mM of β-mercaptoethanol, and 4 mM of magnesium chloride. Moreover, 1.7 mM of MUG in dimethyl sulfoxide was added, except for the assay-specific negative control. The reaction occurred for 1 h at 37 °C in the dark and was finally stopped by the addition of 600 µL 400 mM sodium carbonate solution. A volume of 150 µL of the reaction mixture was transferred to a flat-bottomed black 96-well plate in technical triplicates. The fluorescence (λex/em = 360/465 nm) of the samples was measured on a Tecan Reader infinite 200 PRO (Tecan, Männedorf, Switzerland). The measured values of the negative control without MUG were subtracted as blank values. The values were normalized to the protein content in the lysate supernatant measured by bicinchoninic acid (BCA) assays.

### 2.12. Bicinchoninic Acid Assay

The working solution for the assay was prepared from 50 parts BCA and one part 4% copper (II) sulfate solution. An amount of 200 µL of working solution was mixed with 25 µL of protein sample in a 96-well flat-bottomed plate. A BSA standard series from 0 to 1000 ng/mL was included. The samples were applied as technical duplicates. After sealing the plate with parafilm, the reaction took place at 37 °C for 30 min. The absorbance of the samples was measured at 562 nm in a Tecan Reader infinite 200 PRO (Tecan, Männedorf, CE).

### 2.13. ENPP1 Activity Assay

Thymidine 5′-monophosphate-nitrophenyl ester is converted by the ENPP1 to p-nitrophenol. The latter can be determined photometrically [34]. The medium of the cultured cells was removed and a wash step with PBS was performed. The medium was supplemented with 1 mg/mL of thymidine 5′-monophosphate-nitrophenyl ester sodium salt and applied to the cells. Under these conditions, the cells were incubated for 1 h at 37 °C. Afterwards, the cell culture supernatant was transferred to a 96-well flat-bottomed plate in technical duplicates. The absorbance of the samples was measured at 415 nm using the Tecan Reader infinite 200 PRO (Tecan, Männedorf, CE). Cells were lysed and DNA quantified to normalize the readings.

### 2.14. Statistical Analysis

Data regarding the mRNA expression and protein content in the supernatant were presented as mean ± standard error (SEM). Graphs depicting the quantification of the immunofluorescent labeling were presented as dot plots with additional marking of the mean ± SEM. Data concerning senescence quantification and ENPP1 activity were presented as box and whiskers with bars marking the 10–90 percentile. Non-parametric two-tailed Mann–Whitney U tests were carried out to evaluate the statistical significance levels. Probability (*p*) values equal to 0.05 or below were assumed to be statistically significant.

## 3. Results

### 3.1. CRISPR–Cas9-Mediated Knock-Out in Immortalized Hepatocytes

The transfection of HepIms with a plasmid-based CRISPR–Cas9 system was performed with the FuGene6 transfection reagent followed by the fluorescence-activated cell sorting of transfected cells. Scattered cells were singulated and subsequently analyzed genetically. Single clones were sequenced according to Sanger in the region of the gRNA in *ABCC6* exon 12. Following the TA cloning, sequencing was performed to identify any mutations in single alleles. In addition, a T7 assay was performed to confirm mutagenesis. Subsequently, the *hABCC6* gene expression was determined by the qRT-PCR. The results of the gene-editing approach are shown in Figure 1.

The sequence analysis by Sanger sequencing in the wild-type clones matched the reference sequence (NG_007558.3). By contrast, the base calling for sequences in exon 12 of the *ABCC6*-deficient HepIms was erroneous. Heterozygous mutations in the sequences were noticed, starting three bases upstream of the protospacer adjacent motif (PAM) for clone one and nineteen bases downstream of the PAM for clone 2. In the latter case, the PAM was unidentifiable due to mutagenesis (Figure 1A).

Subsequently, TA cloning was applied to clarify the specific mutations on each allele. It revealed either a wild-type allele or a 10 bp deletion (c.1499_1508del) for clone 1. This deletion resulted in a reading frame shift and, consequently, a premature stop codon leading to a truncation of the protein length from 1503 to 559 amino acids (p.*Thr*500*Thrfs**60). Clone 2 was characterized by either a 31 bp deletion (c.1495_1525del) or a 37 bp deletion (c.1497_1533del). In both cases, these mutations evoked a reading frame shift leading to a protein truncated to 551 amino acids (p.*Lys*499*Profs**54) or 549 amino acids (p.*Lys*499*Lysfs**52), respectively. Regarding these results, clone 1 was defined as heterozygous (*ht^ABCC6^*HepIm) and clone 2 as compound heterozygous (*cht^ABCC6^*HepIm). *wt^ABCC6^*HepIm refers to wild-type single-cell clones (Figure 1B).

Mutagenesis was confirmed by the performance of a T7 assay, which identifies base pair mismatches within the heteroduplexes of amplificates. The PCR products for the primer system used comprised 553 bp when applied on the genomic *hABCC6* gene. The gRNA is located 204 bp downstream of the forward primer, and the PAM can be found 227 bp upstream of the reverse primer. The gel electrophoresis of all amplificates, which were excluded from the incubation with T7 endonuclease, revealed a product longer than 500 bp, the wild-type fragment. This product is detectable in the *wt^ABCC6^*HepIm amplificates treated with endonuclease as well, without the occurrence of further fragments. In addition to the wild-type fragment, further products of approximately 331 and 242 bp were detected by the gel electrophoresis after the treatment with T7 endonucleases in amplificates derived from *ABCC6*-deficient HepIms (Figure 1C).

The *hABCC6* mRNA was significantly reduced to 25.1% (±4.1%) in the *ht^ABCC6^*HepIm and to 3.0% (±0.7%) in the *cht^ABCC6^*HepIm compared to the *wt^ABCC6^*HepIm (Figure 1D).

Undesirable genome editing in genes containing sequences similar to the gRNA was ruled out based on sequencing in the most likely off-targets genes coding for human glutamate ionotropic receptor NMDA type subunit 1 (*hGRIN1*) and human high mobility group nucleosome binding domain 5 (*hHMGN5*), as selected by in silico analysis (Supplement, Appendix A).

### 3.2. First Insight into ABCC6-Deficient Hepatocytes

The *ABCC6*-deficient HepIm clones generated were subjected to characterization for an initial assessment of the impact on their cellular biochemistry. Accordingly, selected targets of the PXE-affected pathobiochemical mechanisms were analyzed.

A first overview of the lipid status was provided by the quantification of the cholesterol and TG in the cell culture supernatant of HepIms cultured for 72 h in a medium containing 10% dFCS. The *wt^ABCC6^*HepIm showed a cholesterol concentration in the supernatant of 0.68 µM (±0.02 µM). This concentration was not significantly different for the *ht^ABCC6^*HepIm, having 0.66 µM of (±0.03 µM) cholesterol in their supernatant. A 10.9% (±4.8%; *p* = 0.042) lower cholesterol concentration of 0.61 µM (±0.03 µM) was detected in the cell culture supernatant for the *cht^ABCC6^*HepIm. Regarding the TG, the concentration in the cell culture supernatant of the *wt^ABCC6^*HepIm (39.67 ± 0.88 µM) was not different from that of the *ht^ABCC6^*HepIm (38.87 ± 1.58 µM), but was significantly higher in the *cht^ABCC6^*HepIm (47.67 ± 1.74 µM). The increase was 20.2% (±5.1%; *p* = 0.003).

A gene expression analysis was performed for *hCYP27A1*, mirroring the bile acid metabolism, *hSCARB1* and *hCD36* representative of the HDL and fatty acid uptake, and *hLDLR* and *hPCSK9* to display the LDL uptake. In addition, the concentration of PCSK9 in cell culture supernatants was determined by an ELISA. Figure 2 presents the data relating to the parameter analysis of the lipid homeostasis.

The gene expression of *hCYP27A1* was reduced to 63.8% (±7.1%) in the *ht^ABCC6^*HepIm and to 49.1% (±4.7%) in the *cht^ABCC6^*HepIm compared to the *wt^ABCC6^*HepIm (Figure 2A). Regarding the gene expression of *hSCARB1*, this parameter was decreased to 64.0% (±4.6%) in the *ht^ABCC6^*HepIm and 71.5% (±4.3%) in the *cht^ABCC6^*HepIm compared to the *wt^ABCC6^*HepIm (Figure 2B). The *hCD36* mRNA was not detectable in either the *ht^ABCC6^*HepIm or the *cht^ABCC6^*HepIm, in contrast to the *wt^ABCC6^*HepIm (Figure 2C). Relative to the *wt^ABCC6^*HepIm, the *hLDLR* mRNA expression was reduced to 60.2% (±4.6%) in the *ht^ABCC6^*HepIm and 73.8% (±3.9%) in the *cht^ABCC6^*HepIm (Figure 2D). While no significant difference in the *hPCSK9* gene expression could be identified in the *ht^ABCC6^*HepIm compared to the *wt^ABCC6^*HepIm, a reduced gene expression to 23.1% (2.5%) was present in the *cht^ABCC6^*HepIm (Figure 2E). However, the PCSK9 concentration in the cell culture supernatant was reduced to 8.3% (±1.0%) not only for the *cht^ABCC6^*HepIm but also for the *ht^ABCC6^*HepIm, whose PCSK9 concentration in the supernatant corresponded to 30.3% (±4.0%) of those from the *wt^ABCC6^*HepIm (Figure 2F).

The gene expression pattern of other ABC transporters was representatively evaluated by the determination of the *hABCA1*, *hABCC2,* and *hABCC3* mRNA expression via a qRT-PCR and displayed in Figure 3.

The *ABCC6* deficiency showed no significant effect on the *hABCA1* expression in the HepIm (Figure 3A). Furthermore, the *hABCC2* gene expression was unchanged in the *ht^ABCC6^*HepIm compared to the *wt^ABCC6^*HepIm. However, the *cht^ABCC6^*HepIm exhibited a 1.8-fold (±0.2) higher mRNA expression of this transporter than the *wt^ABCC6^*HepIm (Figure 3B). The *ht^ABCC6^*HepIm and *cht^ABCC6^*HepIm *hABCC3* gene expression decreased to 68.0% (±5.9%) and 40.1% (±5.3%), respectively, in both clones compared to the expression in the *wt^ABCC6^*HepIm (Figure 3C).

The remodeling of the ECM was assessed by the gene expression via the qRT-PCR and protein detection via immunofluorescence for COL1A1 and DCN. The results are illustrated in Figure 4.

The relative *hCOL1A1* mRNA expression decreased to 1.8% (±0.4%) in the *ht^ABCC6^*HepIm and 4.5% (±1.1%) in the *cht^ABCC6^*HepIm compared to that in the *wt^ABCC6^*HepIm (Figure 4A). The COL1A1 protein expression analyzed at the fluorescence level was detected to be only 36.7% (±2.7%, *ht^ABCC6^*HepIm) and 35.7% (±2.7%, *cht^ABCC6^*HepIm) in relation to that of the *wt^ABCC6^*HepIm, respectively (Figure 4B,C, representative).

The *hDCN* gene expression was significantly reduced in the *ht^ABCC6^*HepIm (5.6 ± 1.0%) and *cht^ABCC6^*HepIm (1.9 ± 0.3%) relative to the *wt^ABCC6^*HepIm (Figure 4D). These data were reflected in the quantification of the immunofluorescence-based DCN detection as a decrease to 48.5% (±3.2%) in the *ht^ABCC6^*HepIm and 51.1% (±2.8%) in the *cht^ABCC6^*HepIm compared to the *wt^ABCC6^*HepIm (Figure 4E,F, representative).

Additionally, the mRNA expression of *hFBN1* and *hFBN2*, representing the assembly of elastic fibers, as well as of the tissue inhibitor of MMP 1 (*hTIMP1*) and *hMMP2*, mirroring the proteolytic potential, was measured via the qRT-PCR. The quantitative gene expressions are provided in Figure 5.

The *ABCC6* deficiency in the *ht^ABCC6^*HepIm and *cht^ABCC6^*HepIm resulted in an *hFBN1* gene expression level diminished to 61.8% (±4.2%) and 32.0% (±3.0%), respectively, compared to the *wt^ABCC6^*HepIm (Figure 5A). On the contrary, the *ABCC6* deficiency of the *cht^ABCC6^*HepIm resulted in a significant 15.1-fold (±4.0) induction of the *hFBN2* mRNA expression relative to the *wt^ABCC6^*HepIm, while there was no significant change in the *hFBN2* gene expression in the *ht^ABCC6^*HepIm (Figure 5B). Regarding the *hTIMP1* gene expression, there was a decrease to 76.9% (±5.9%) in the *ht^ABCC6^*HepIm and 40.3% (±4.1%) in the *cht^ABCC6^*HepIm relative to the *wt^ABCC6^*HepIm (Figure 5C). The *ht^ABCC6^*HepIm showed a reduction in the hMMP2 gene expression to 79.0% (±5.9%), whereas the *cht^ABCC6^*HepIm presented an induction to 1.4-fold (±0.1) compared to the *wt^ABCC6^*HepIm (Figure 5D).

We further evaluated the calcification inhibition potential of *ABCC6*-deficient hepatocytes by the gene expression analysis of *hMGP* and *hENPP1* and an ENPP1 activity assay. The presentation of the results can be found in Figure 6.

The *ht^ABCC6^*HepIm showed an 8.7-fold (±0.9) and the *cht^ABCC6^*HepIm a 5.2-fold (±0.5) induction of the *hMGP* mRNA expression relative to the *wt^ABCC6^*HepIm (Figure 6A). In the case of *hENPP1*, gene expression levels were found to be 2.6-fold (±0.2) higher in the *ht^ABCC6^*HepIm and 2.5-fold (±0.3) higher in the *cht^ABCC6^*HepIm than in the *wt^ABCC6^*HepIm (Figure 6B). The *ABCC6* deficiency of the *cht^ABCC6^*HepIm resulted in an increased ENPP1 activity compared to the *wt^ABCC6^*HepIm. This corresponded to a 1.4-fold (±0.1) increase after culturing for 3 d and a 3.7-fold (±0.3) increase after 21 d of culturing (Figure 6C).

The senescent status of *ABCC6*-deficient hepatocytes was investigated representatively by quantifying the p21 gene and protein expression, *hIL6* and *hIL1b* gene expression, and IL6 content in the cell culture supernatant. In addition, the activity of SA-β-galactosidase was quantified by the conversion of the substrate MUG. Figure 7 summarizes the data on the senescence status of the hepatocytes.

The relative *hCDKN1A* mRNA expression was equivalent to 58.1% (±10.8%) in the *ht^ABCC6^*HepIm and 39.0% (±6.4%) in the *cht^ABCC6^*HepIm compared to that in the *wt^ABCC6^*HepIm (Figure 7A). This was presented at the protein expression level as a mean nuclear fluorescence intensity after the p21 labeling of 44.7% (±2.9%) in the *ht^ABCC6^*HepIm and 38.5% (±2.0%) in the *cht^ABCC6^*HepIm (Figure 7B) compared to the *wt^ABCC6^*HepIm. The images shown present a representative picture (Figure 7C).

The *hIL1b* mRNA expression corresponded to 24.1% (±1.9%) in the *ht^ABCC6^*HepIm and 41.8% (±4.0%) in the *cht^ABCC6^*HepIm relative to the *wt^ABCC6^*HepIm. While the *cht^ABCC6^*HepIm had a significantly lower *hIL6* gene expression (42.0% ± 4.0%), no significant difference in expression was detectable between the *ht^ABCC6^*HepIm and *wt^ABCC6^*HepIm (Figure 7E). The quantification of IL6 in the cell culture supernatant of the HepIm showed only a trend but no significant reduction to 73.8% (±8.4%) for the *cht^ABCC6^*HepIm compared to the *wt^ABCC6^*HepIm (Figure 7F).

The relative SA-β-galactosidase activity, represented by the photometrically determinable conversion of its substrate, was reduced to 71.9% (±3.7%) and 76.5% (±3.9%) in the *ht^ABCC6^*HepIm and in the *cht^ABCC6^*HepIm compared to *wt^ABCC6^*HepIm, respectively (Figure 7G).

Furthermore, we selected and analyzed the gene and protein expression of the complement components C1r and C3 as an index of the inflammatory status of *ABCC6-*deficient hepatocytes. The results are depicted in Figure 8.

The comparison of the *hC1R* mRNA expression revealed that it corresponded to 22.0% (±2.1%) in the *ht^ABCC6^*HepIm and 18.6% (±2.6%) in the *cht^ABCC6^*HepIm (Figure 8A) relative to that in the *wt^ABCC6^*HepIm. The *hC3* gene expression represented 18.8% (±1.8%, *ht^ABCC6^*HepIm) and 10.3% (±2.0%, *cht^ABCC6^*HepIm) in the *ABCC6*-deficient HepIm compared to the *wt^ABCC6^*HepIm (Figure 8B).

The C1r concentration in the cell culture supernatants was detected to be 19.8% (±2.9%) for the *ht^ABCC6^*HepIm and 27.0% (±2.4%) for the *cht^ABCC6^*HepIm relative to the *wt^ABCC6^*HepIm (Figure 8C). The C3 concentration in the cell culture supernatant of the *ht^ABCC6^*HepIm dropped to 6.3% (±1.0%) and to 8.1% (±0.7%) in the *cht^ABCC6^*HepIm relative to that in the *wt^ABCC6^*HepIm (Figure 8D).

## 4. Discussion

Mutations in the ABCC6 transporter are causative in the rare disease PXE. Neither the substrate nor the pathobiochemistry leading to the symptoms of PXE are fully understood [35,36]. The expression pattern of ABCC6 shows tissue-specific differences. Studies in humans and mice have shown that the transporter is expressed predominantly in the liver [37,38]. However, it is difficult to obtain liver tissue from humans due to ethical reasons and the clinical risks of an invasive biopsy. Therefore, the aim of our work was to establish an *ABCC6*-deficient in vitro hepatocyte system to study the pathobiochemistry of PXE. We successfully generated a heterozygous and compound heterozygous *ABCC6* knock-out clone of the HepIm by the transfection with a plasmid-based *hABCC6*-specific CRISPR–Cas9 system, fluorescence selection, and single-cell clonal expansion. Both clones showed mutations in the intended genomic area and a significant reduction in the *hABCC6* gene expression. The gRNA directs the Cas9 to a site in exon 12 just before the first nucleotide binding domain. These domains, of which ABCC6 contains two, are thought to be essential for the transporter’s function, as most mutations in PXE patients are described in the coding sequence of these regions [39]. Based on the position of the mutations in exon 12 and the resulting termination of the translation shortly after the described mutations (stop codon within exon 12, predicting no translation of NBD_1_, NBD_2_, and TMD_2_), it can be assumed that this leads to a loss of function of ABCC6.

Limitations need to be considered when evaluating this cell culture model.

A confirmation of the efficient depletion of the human ABCC6 protein could not be confirmed via the Western blot or immunocytochemistry due to problems with unspecific antibody binding, potentially cross-reacting with other ABC transporters. Moreover, due to missing knowledge about the ABCC6 substrate, activity assays are not applicable. Although hepatocytes form the parenchymal part of the liver and represent about 80% of the biomass, the tissue in vivo contains other cell types, such as Kupffer cells, is permeated by fine blood vessels, and is embedded in the ECM. These factors may influence the phenotype when transferring in vitro results to in vivo systems [40,41]. Further research may, therefore, concentrate on extending our model in terms of three-dimensional applications, for example, in microfluidic systems and complex co-cultures with different liver cell subtypes [30,42,43].

Regarding several PXE pathobiochemical markers, the *ht^ABCC6^*HepIm exhibited an expression pattern comparable to those in either the *wt^ABCC6^*HepIm (e.g., *hABCC2*, *hFBN2*, and *hIL6*) or *cht^ABCC6^*HepIm (e.g., *hLDLR*, *hCOL1A1*, *hC3*) and, in parts, an intermediate expression (e.g., hCYP27A1, *hFBN1*, and *hCDKN1A*). Therefore, the heterozygous model is particularly suitable for future studies to gain a deeper understanding of the dosage effect of the partial loss of ABCC6 in PXE, representing carriers of a heterozygous *hABCC6* mutation which often shows a milder phenotype [44]. Due to the largely congruent trends in the gene expression in the *ht^ABCC6^*HepIm and *cht^ABCC6^*HepIm, the results below are classified as a comparison between the *ABCC6*-deficient and wild-type HepIm.

Although the present study cannot, and was not intended to, fully characterize the generated clones, the first analysis performed shows how this in vitro model fits into the pathobiochemical model of the PXE disease. The cell culture supernatants of the *ABCC6*-deficient HepIm showed an increased concentration of TGs and a decreased concentration of cholesterol. Although it is not clear at this stage whether these differences are due to changes in the influx or efflux, they indicate a disturbance in lipid metabolism. A reduced TG utilization in the context of the fatty acid internalization by CD36 is implied by the lack of gene expression of this receptor on the *ABCC6*-deficient HepIm. In the mouse model, a lower level of expression of this receptor has already been described in the liver tissue of *Abcc6^-/-^* animals compared to wild-type animals [45]. It is possible that this lack of the TG influx into the liver leads to the elevated serum TG levels observed in *Abcc6^-/-^* mice and PXE patients [45,46]. The uptake of cholesterol-rich LDL particles by the LDLR and the inhibition of this process by PCSK9 is also affected by the *ABCC6* deficiency in the HepIm. However, while PCSK9 levels are elevated in both peripheral cell culture supernatants and patient sera, here, they are reduced in the *ABCC6*-deficient HepIm [16,45]. A cholesterol flux assay could provide a deeper insight into the management of the cholesterol content and metabolic rate of the HepIm and, thus, its involvement in the systemic cholesterol excess. Furthermore, it is important to consider that the cellular cholesterol balance depends on reverse cholesterol transport from the periphery as well as the metabolization to bile acids. *ABCC6*-deficient human adipocytes, for example, show a reduced *hABCA1* gene expression, which indicates a reduced reverse cholesterol transport and might be associated with the reduced gene expression of *hSCARB1*, the receptor of HDL in the reverse cholesterol transport, in HepIms observed in our study [17]. Moreover, both the reduced gene expression of *hCYP27A1*, representative of bile acid metabolism, and the altered gene expression profile of the apical ABC transporters, C2 and C3, indicate an altered bile acid metabolite profile in the *ABCC6*-deficient HepIm. The fact that an *ABCC6* knock-out leads to changes in the expression pattern of various ABC transporters has already been demonstrated in the mouse model and is confirmed here [22]. Further studies on the ABC transporter expression and the analysis of the bile acid metabolite profile of HepIms could be the subject of future research.

The loss of liver function due to the fibrogenesis of this organ has not been studied for PXE patients so far. The analysis of HepIms for collagen and DCN, a PG involved in the assembly of collagen fibers, showed a strongly reduced expression in *ABCC6*-deficient cells associated with an anti-fibrotic effect [15,47]. This contrasts with the overexpression of profibrotic factors, including DCN, in PXE fibroblasts [48]. Considering FBN1, the predominant isoform of fibrillin in adult organisms, deficiencies occur in Marfan syndrome and lead to the incomplete assembly of elastic fibers. In a mouse model, the knock-out of *Fbn1* leads to the induction of the *Fbn2* gene expression, which did not rescue the phenotype [11,49]. The downregulation of *FBN1* and the compensatory overexpression of *FBN2* in HepIms, therefore, indicate deficits in the elastic fiber assembly. Furthermore, reduced *TIMP1* and induced *MMP2* gene expressions indicated an increased ECM degradation potential. This has also been observed in PXE fibroblasts [13]. Regarding the ECM, the *ABCC6*-deficient HepIm therefore showed an ambivalent picture. While fibrotic changes due to increased collagen deposition, as described in PXE fibroblasts, were not observed in this cell type, HepIm and PXE fibroblasts with an *ABCC6* deficiency appeared similar regarding their limited assembly of elastic fibers and the elevated proteolytic potential by MMPs.

The examination of the calcification regulation factors, *MGP* and *ENPP1,* revealed an induced gene expression and increased ENPP1 activity in the *ABCC6*-deficient HepIm. The PXE fibroblasts also show an induced MGP expression, whereas no increased *MGP* gene expression is detectable in the mouse liver tissue. Interestingly, MGP is largely uncarboxylated in both systems and, thus, inactive [10,50]. For HepIms, it is, therefore, necessary to investigate the functionality of the MGP to make further statements about the inhibition of calcification. As far as ENPP1 is concerned, PXE fibroblasts show both a reduced *ENPP1* gene expression and a lower extracellular PP_i_ concentration [51]. The latter can also be detected in the serum of patients [52]. The increased ENPP1 expression of HepIms thus represents rather anti-calcifying properties, suggesting that the ectopic calcification of peripheral tissues represented by fibroblast cell cultures might be influenced mainly by local mechanisms. However, the *ABCC6*-deficient HepIm might try to compensate for the PP_i_ deficiency by increasing the ENPP1 expression and activity.

The results for the senescence-associated and inflammatory parameters, including SA-β-galactosidase activity and p21, IL1b, IL6, C1R, and C3 expression, indicate a less senescent and anti-inflammatory phenotype in the *ABCC6*-deficient HepIm compared to the wild-type HepIm. This is in contrast to observations in human fibroblasts, which show a p21-dependent senescence-associated phenotype [24]. A special focus of future studies should be placed on the connection between the remodeling of the ECM and the dysregulation of the inflammatory response, which plays a decisive role in end-stage fibrosis. In particular, cocultivation models between hepatocytes and fibroblasts could provide helpful insights into the effects of ABCC6 deficiency on systemic fibrotic processes. Furthermore, Miglionico et al. developed an *ABCC6* knock-down system in HepG2 which, in contrast to our HepIm, showed an increased SA-β-galactosidase activity and p21 expression and was, thus, closer to the fibroblast phenotype. However, the results in fibroblasts and HepG2 are not completely consistent, as the *ABCC6* knock-down HepG2 did not show increased oxidative stress [53]. This comparison highlights a limitation not only of our HepIm model but also of the use of cell lines in general. While HepIms may be less sensitive to senescence stimuli due to their immortalization, carcinoma cell lines, such as HepG2, differ significantly from healthy cells in their secretion of proinflammatory cytokines [54]. The observation and comparison of different *ABCC6*-deficient hepatocyte models allows the identification of underlying pathobiochemical mechanisms based on similarities.

Apart from the choice of the cell culture model (cell lines or immortalized primary cells) further limitations of our model must be taken into account. Also discussed above was the need to increase the number of clones to confirm our results. The latter was limited due to the low transfection and gene editing efficiency. CRISPR–Cas systems are subject to constant change, which allows for the adaptation of protocols to increase efficiencies. Our model should not be seen as a rigid template, but as a dynamic starting point. And finally, some aspects of the PXE pathobiochemistry have been outlined in our study, but each of them can and should be clarified by more in-depth analyses. Besides the described limitations, we established a new and promising human cell culture model, based on human hepatocytes representing the main expression site of ABCC6. This ABCC6-deficient hepatocyte model might also be suitable for uncovering the in vivo substrate of ABCC6, e.g., by metabolomic profiling approaches.

## 5. Conclusions

Our initial results are a proof of concept for our *ABCC6*-deficient hepatocytes model, which resemble impairments in biochemical processes described for PXE before. They point towards an influence of *ABCC6* deficiency regarding the membrane transfer and the metabolism of various lipids leading to the lipid metabolism deficits observed in patients and mice. Moreover, defects in the ECM remodeling in parts mimicking those observed in fibroblasts were identified. A first impression of an anti-calcifying phenotype might be supported by future studies on calcification in *ABCC6*-deficient HepIm clones. Special caution must be taken when interpreting the results of an anti-inflammatory and less senescent phenotype in the *ABCC6*-deficient HepIm, which requires the addition of further model systems and comparative studies. Furthermore, observations made in the present study might be consolidated by the expansion of the model, for example, by running a rescue approach or an increase in the number of clones. In particular, the ability to analyze clones with single nucleotide missense mutations may more accurately mimic the genetic defects of PXE.

We emphasize that the capability of this model is not yet exhausted and that our analysis was performed to provide a first impression of the phenotype of these *ABCC6*-deficient hepatocyte clones. Applying our approach to generate a wide range of clones with mutations in different regions of the gene may help us to understand the relationship between the phenotype and genotype. Moreover, detailed pathobiochemical analyses of our newly generated in vitro model will follow and help one to understand the hepatocytic involvement in the pathogenesis of PXE.

## Figures and Tables

**Figure 1 cells-14-00576-f001:**
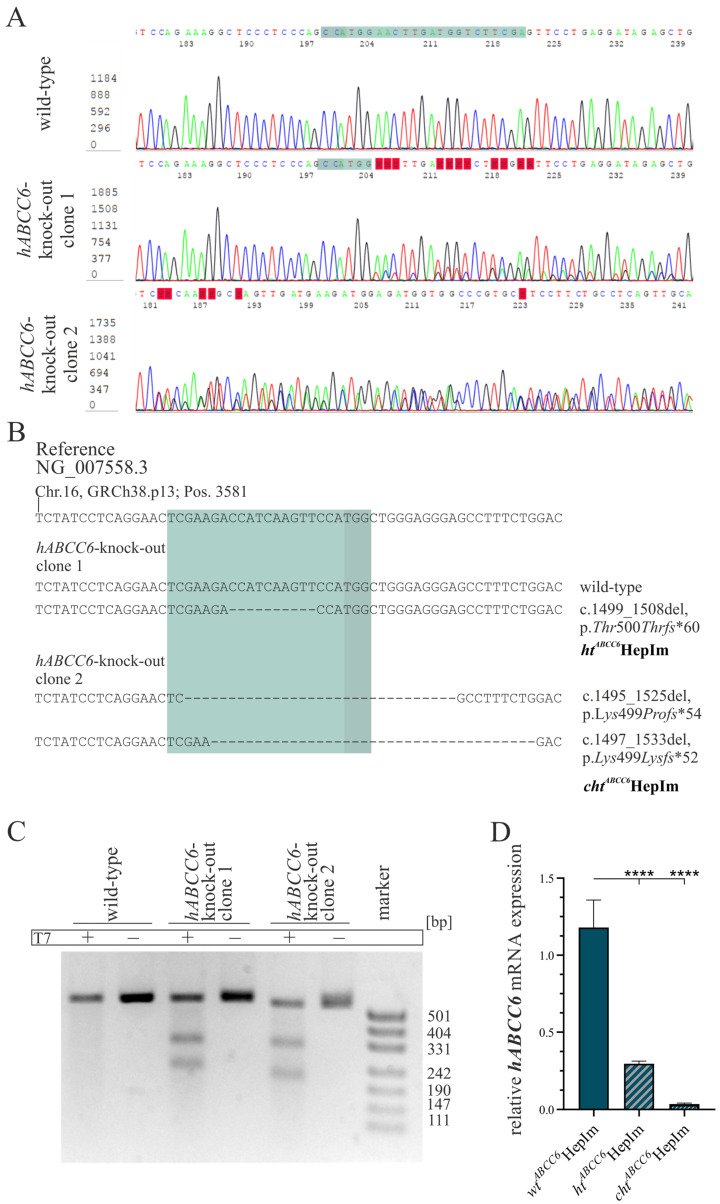
Plasmid-based CRISPR–Cas9 *hABCC6* knock-out in HepIm. (**A**) Sanger sequencing section of guide sequence region of *hABCC6* exon 12 (green) and protospacer adjacent motif (PAM) (dark green) was performed for wild-type and two *hABCC6* knock-out HepIm single clones. (**B**) Sequences of single clones resulting from TA cloning in guide sequence region (green) resulted in clones showing one heterozygous (ht) and two compound heterozygous (cht) deletions of 10 to 37 bp. (**C**) *hABCC6* exon 12 amplificates from wild-type and *hABCC6* knock-out HepIm clones were separated via agarose gel electrophoresis after incubation in presence or absence of endonuclease T7. (**D**) *hABCC6* mRNA expression in wild-type (*wt*; *n* = 2), heterozygous (*ht*, *n* = 1), and compound heterozygous (*cht*; *n* = 1) HepIm was determined by qRT-PCR. Data are presented as means ± standard errors. *p* < 0.0001 (****).

**Figure 2 cells-14-00576-f002:**
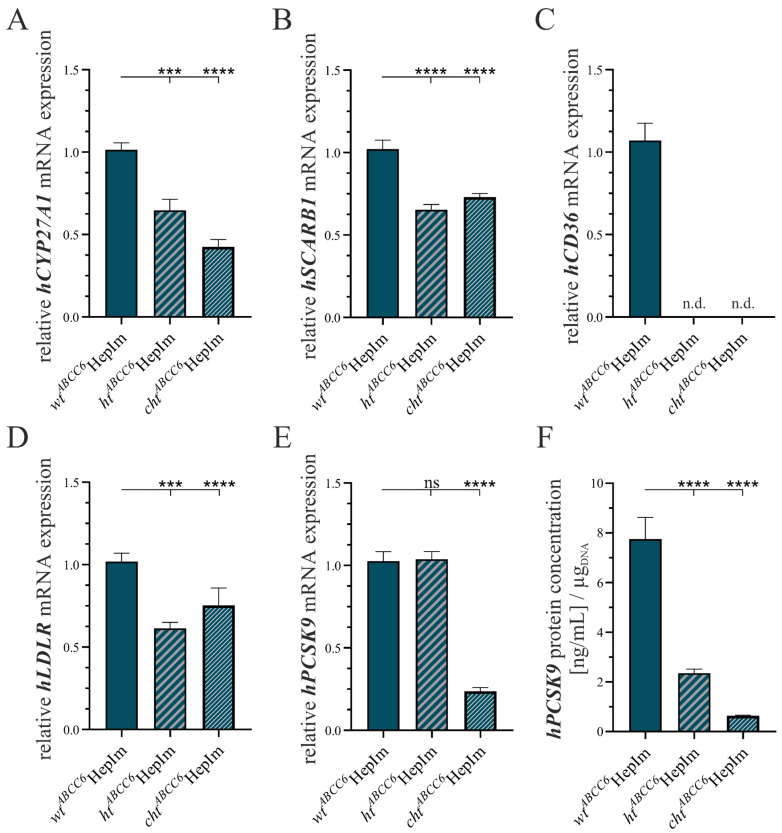
Expression analysis of selected markers of lipid homeostasis in *wt^ABCC6^*HepIm (*n* = 2), *ht^ABCC6^*HepIm (*n* = 1), and *cht^ABCC6^*HepIm (*n* = 1). mRNA expression of (**A**) *hCYP27A1*, (**B**) *hSCARB1*, (**C**) *hCD36*, (**D**) *hLDLR,* and (**E**) *hPCSK9* was determined by qRT-PCR. Relative gene expression below detection limit is labeled as not detectable (n.d.). (**F**) PCSK9 content in cell culture supernatant was measured with ELISA and normalized to DNA content. Data are shown as mean ± SEM. Mann–Whitney U significance test levels: not significant (ns), *p* < 0.001 (***), and *p* < 0.0001 (****).

**Figure 3 cells-14-00576-f003:**
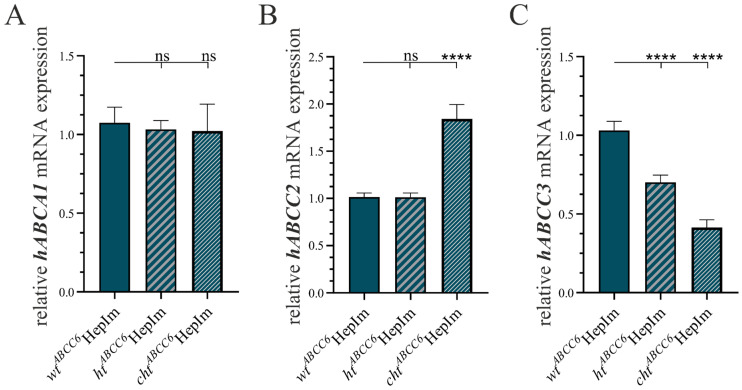
Expression analysis of selected ABC transporter family members in *wt^ABCC6^*HepIm (*n* = 2), *ht^ABCC6^*HepIm (*n* = 1), and *cht^ABCC6^*HepIm (*n* = 1). mRNA expression of (**A**) *hABCA1*, (**B**) *hABCC2,* and (**C**) *hABCC3* was determined by qRT-PCR. Data are shown as mean ± SEM. Mann–Whitney U significance test levels: not significant (ns), *p* < 0.0001 (****).

**Figure 4 cells-14-00576-f004:**
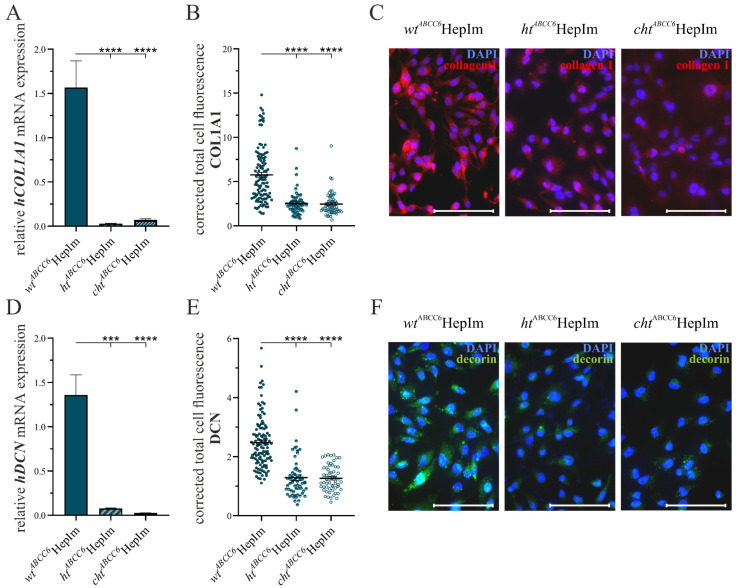
Expression analysis of selected markers of ECM remodeling in *wt^ABCC6^*HepIm (*n* = 2), *ht^ABCC6^*HepIm (*n* = 1), and *cht^ABCC6^*HepIm (*n* = 1). mRNA expression of (**A**) *hCOL1A1* and (**D**) *hDCN* was determined by qRT-PCR. Immunofluorescence-based detection of (**B**) COL1A1 and (**E**) DCN was quantified by corrected total cell fluorescence intensity using ImageJ. Data are shown as mean ± SEM. Mann–Whitney U significance test levels: *p* < 0.001 (***) and *p* < 0.0001 (****). Representative images of immunofluorescence-based labeling of (**C**) COL1A1 and (**F**) DCN are provided. Scale bar: 100 µm.

**Figure 5 cells-14-00576-f005:**
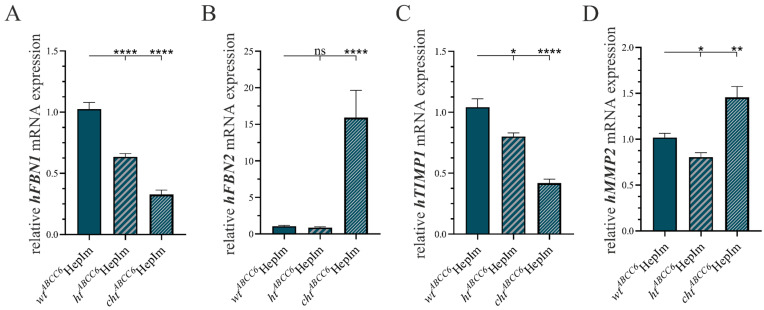
Gene expression analysis of selected markers of ECM remodeling in *wt^ABCC6^*HepIm (*n* = 2), *ht^ABCC6^*HepIm (*n* = 1), and *cht^ABCC6^*HepIm (*n* = 1). mRNA expression of (**A**) *hFBN1*, (**B**) *hFBN2*, (**C**) *hTIMP1,* and (**D**) *hMMP2* was determined by qRT-PCR. Data are shown as mean ± SEM. Mann–Whitney U significance test levels: not significant (ns), *p* < 0.05 (*), *p* < 0.01 (**), and *p* < 0.0001 (****).

**Figure 6 cells-14-00576-f006:**
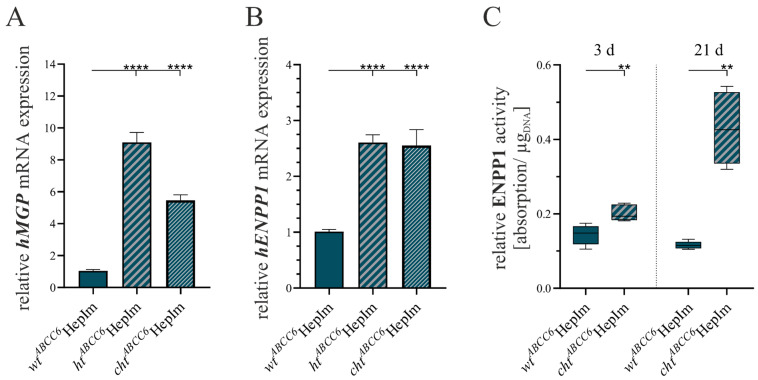
Analysis of selected markers of calcification regulation in *wt^ABCC6^*HepIm (*n* = 2), *ht^ABCC6^*HepIm (*n* = 1), and *cht^ABCC6^*HepIm (*n* = 1). mRNA expression of (**A**) *hMGP* and (**B**) *hENPP1* was determined by qRT-PCR. (**C**) ENPP1 activity was assessed with calorimetric enzyme activity test and normalized to DNA content. Data are shown as mean ± SEM. Mann–Whitney U significance test levels: *p* < 0.01 (**) and *p* < 0.0001 (****).

**Figure 7 cells-14-00576-f007:**
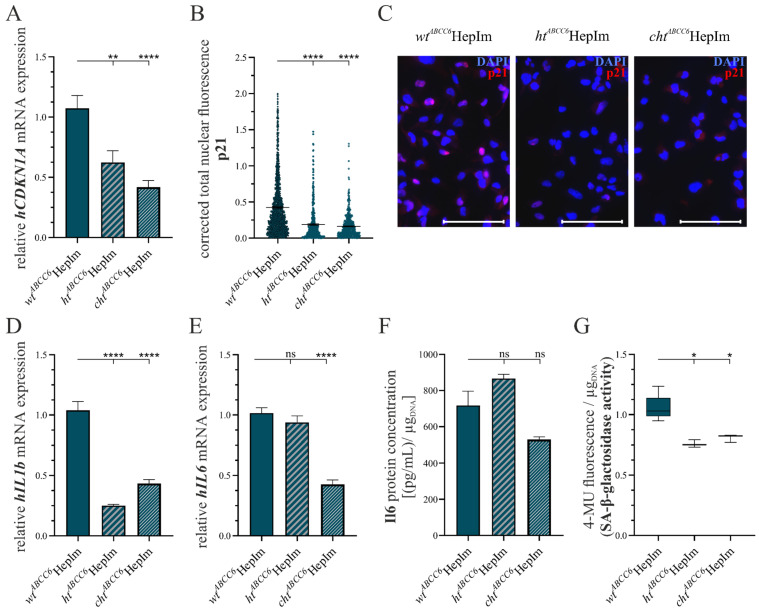
Analysis of selected markers of senescence in *wt^ABCC6^*HepIm (*n* = 2), *ht^ABCC6^*HepIm (*n* = 1), and *cht^ABCC6^*HepIm (*n* = 1). mRNA expression of (**A**) *hCDKN1A*, (**D**) *hIL1b*, and (**E**) *hIL6* was determined by qRT-PCR. (**B**) Immunofluorescence-based detection of p21 was quantified by corrected total cell fluorescence intensity using ImageJ. (**C**) Representative images of immunofluorescence-based detection of p21 are provided. Scale bar: 100 µm. (**F**) IL6 protein concentration was measured in cell culture supernatant and normalized to DNA content. (**G**) SA-β-galactosidase activity was evaluated by fluorometric tracking of substrate conversion and normalized to DNA content. Data are shown as mean ± SEM. Mann–Whitney U significance test levels: not significant (ns), *p* < 0.05 (*), *p* < 0.01 (**), and *p* < 0.0001 (****).

**Figure 8 cells-14-00576-f008:**
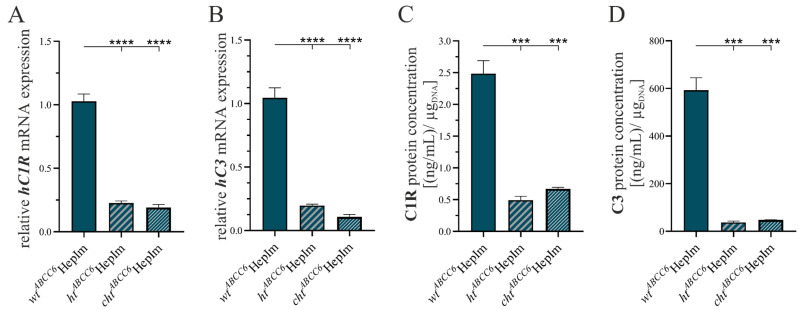
Expression analysis of selected markers of complement system in *wt^ABCC6^*HepIm (*n* = 2), *ht^ABCC6^*HepIm (*n* = 1), and *cht^ABCC6^*HepIm (*n* = 1). mRNA expression of (**A**) *hC1R* and (**B**) *hC3* was determined by qRT-PCR. (**C**) C1R and (**D**) C3 content in cell culture supernatant was measured with ELISA and normalized to DNA content. Data are shown as mean ± SEM. Mann–Whitney U significance test levels: *p* < 0.001 (***) and *p* < 0.0001 (****).

**Table 1 cells-14-00576-t001:** gRNA oligonucleotides for the plasmid-based CRISPR–Cas9 genome editing. The overlaps for cloning into the vector PX558 are underlined. The oligonucleotides were obtained from Biomers (Ulm, Germany).

Gene	5′-3′ Primer Sequences	PAM
*hABCC6* (exon 12)	CACCGTCGAAGACCATCAAGTTCCA	TGG
	AAACTGGAACTTGATGGTCTTCGAC	

**Table 2 cells-14-00576-t002:** Primer systems for amplification of genomic DNA. T_A_ indicates annealing temperature in °C. Product size is provided in bp. Oligonucleotides were purchased from Biomers (Ulm, Germany).

Gene	5′-3′ Primer Sequences	T_A_	Product Size
*hABCC6*	CTGTTCTCCGGGCATCAGAGGATGGACGGGGTGGTAGGAT	59	554
*hGRIN1*	GCGCCGCTAACCATAAACAAGAATCTCCTGCGGAGGGACG	56	213
*hHMGN5*	AGCAGATGCTTGTGCCAGTTCCCCACCCAAGGGGTTTAC	56	218

**Table 3 cells-14-00576-t003:** Sequences, annealing temperatures (T_A_ in °C), efficiencies, and resulting product sizes (in bp) of primers used for qRT-PCR.

Gene	Primer Sequence (5′-3′)	T_A_	Efficiency
*hABCA1*	ATCCCCAGCACAGCCTAT	60	1.88
	TCTCCCCAAACCTTTCCA		
*hABCC2*	CACAAGCAACTGCTGAAC	60	1.76
	TGCCAAGAGGAATGACGA		
*hABCC3*	GCTCCAGCTTCCTCATCA	60	1.78
	TGGAGCACAGGAACATCA		
*hABCC6*	CCTGCTGATGTACGCCTT	60	1.92
	ACGCGAGCATTGTTCTGA		
*hC1R*	GAGGAGAATGCCCAGTGGTGG	68	1.92
	GCTTCACCCTGTATCCCGTG		
*hC3*	GAGAAGACTGTGCTGACCCC	68	1.91
	GATGCCTTCCGGGTTCTCAA		
*hCD36*	CCTGCTTATCCAGAAGAC	59	2.00
	CACAGCCAGATTGAGAAC		
*hCDKN1A1*	GCAGACCAGCATGACAGATTTC	66	1.81
	ACCTCCGGGAGAGAGGAAAA		
*hCOL1A1*	GATGTGCCACTCTGACT	63	1.74
	GGGTTCTTGCTGATG		
*hCYP27A1*	CAAACTCCCGGATCATAG	59	2.00
	GACCACCTTGTACTTCTG		
*hDCN*	CCTTCCGCTGTCAATG	63	1.76
	GCAGGTCTAGCAGAGTTG		
*hENPP1*	AATGCCCCTTTGGACATC	59	1.72
	CCCGTAACTCACTTTGGT		
*hFBN1*	TCCCGTGGGATATGTGCTCAG	61	1.97
	ACAGCCTTCTCCATCAGGTCTC		
*hFBN2*	AACACGCCAGGAAGTTACAG	63	2.00
	ATCTAGTTCACACCGCTCAC		
*hGAPDH*	AGGTCGGAGTCAACGGAT	63	1.84
	TCCTGGAAGATGGTGATG		
*hHMBS*	CTGCCAGAGAAGAGTGTG	63	1.92
	AGCTGTTGCCAGGATGAT		
*hIL1B*	ACAGATGAAGTGCTCCTTCCA	63	1.94
	GTCGGAGATTCGTAGCTGGAT		
*hIL6*	ACAGCCACTCACCTCTTCAG	63	1.91
	GTGCCTCTTTGCTGCTTTCAC		
*hLDLR*	CGACTGCAAGGACAAATCTG	61	2.00
	AGTCATATTCCCGGTCACAC		
*hMGP*	AGCGGTAGTAACTTTGTG	63	1.97
	GTGGACAGGCTTAGAG		
*hPCSK9*	CAGCCTGGTGGAGGTGTATC	62	1.70
	CCTATGAGGGTGCCGCTAAC		
*hRPL13*	CGGAAGGTGGTGGTCGTA	63	1.87
	CTCGGGAAGGGTTGGTGT		
*hSCARB1*	TGGGCTCTTCACGGTGTTC	63	2.00
	CCTTCGTTGGGTGGGTAGATG		
*hSDHA*	AACTCGCTCTTGGACCTG	63	1.93
	GAGTCGCAGTTCCGATGT		

## Data Availability

The original raw data and materials presented in this study will be made available upon request. Further inquiries can be directed to the corresponding author.

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
