# Peer review of "Investigation on ABCC6-Deficient Human Hepatocytes Generated by CRISPR–Cas9 Genome Editing"

_cells, 2025, doi:10.3390/cells14080576_

Round 1
Reviewer 1 Report
Comments and Suggestions for Authors
Plumers et al describe the generation of ABCC6 knock-out Hepatocyte cell line which mimics the metabolic characteristics of the PXE in vitro using CRISPR/Cas9.
However, the majority of ABCC6 mutations in PXE are missense variants that impair protein function rather than causing complete loss of function, it remains uncertain whether the heterozygous or compound heterozygous cell lines generated in this study fully or properly recapitulate the pathophysiology of PXE. A more physiologically relevant model might involve the generation of induced pluripotent stem cells (iPSCs) from PXE fibroblasts, which could then be differentiated into hepatocyte-like cells and hepatocyte-like organoids. This approach would better preserve patient-specific genetic backgrounds and disease-relevant mutations rather than relying on a commercially available immortalized hepatocyte cell line.
Minor Concern
- The authors used a plasmid-based system for gene editing instead of a more efficient ribonucleoprotein (RNP) or mRNA-based approach. It is unclear why this method was chosen, especially given the lower editing efficiency with plasmid transfection. The study should provide a rationale for this choice and discuss any associated limitations.
Major Concerns
- There is no information on the gene-editing efficiency of CRISPR-Cas9 in transfected cells. The authors should provide a quantitative assessment of editing efficiency, including indel frequencies in bulk populations before clone selection. A figure and detailed result section highlighting this data would significantly strengthen the study.
- While gene expression data is presented, it is essential to confirm the complete or partial loss of ABCC6 protein in edited hepatocytes. Western blot and immunocytochemistry should be performed in both bulk-transfected cells and selected clones to demonstrate the absence of ABCC6 protein expression.
- Additional gene-editing experiments targeting exons harboring frequent mutations versus non-mutated regions would provide stronger evidence that the cellular model truly reflects PXE.
- Only two clones were selected for characterization of the PXE phenotype. The authors should justify this choice and ideally expand the analysis to multiple independent clones. High-throughput sequencing or TA cloning could be used to characterize the full spectrum of mutations introduced by CRISPR-Cas9, ensuring a more comprehensive mutational landscape. Additionally, single-nucleotide missense mutations would be more appropriate to mimic actual PXE-associated genetic defects rather than large deletions.
- Transmembrane domain prediction for selected KO or Ki clones should be performed and potential impact of the mutagenesis should be discussed
- Given that ectopic calcification is the hallmark of PXE, the study must include a calcification assay. While the authors mention that this will be performed in the future, preliminary calcification assays should be conducted to demonstrate whether ABCC6-deficient hepatocytes exhibit pro-calcification phenotypes. Even if no significant changes are observed, reporting this data would be valuable for understanding the role of hepatocytes in systemic mineralization.
Reviewer 2 Report
Comments and Suggestions for Authors
The manuscript entitled ‘A Hepatocyte CRISPR Cas9 Knock-Out Model for Pseudoxanthoma Elasticum Pathobiochemistry Investigation’ by Ricarda Plümers et al., provides data on an in vitro model for Pseudoxanthoma elasticum (PXE) disease. In humans, this disease shows ocular, dermal, and vascular calcification of elastic fibers due to mutations in the ATP-binding cassette transporter subfamily C member 6 (ABCC6). As ABCC6 is mainly expressed in liver tissue, the authors developed an ABCC6 genome editing system for human immortalized hepatocytes (HepIm) to address this hypothesis. From the methodological perspective, the HepIm were transfected with an ABCC6-specific clustered regulatory interspaced short palindromic repeat (CRISPR-Cas9) genome editing plasmid, where a heterozygous (htABCC6HepIm) and a compound heterozygous (chtABCC6HepIm) clone of HepIm was identified. The ABCC6 knock-out model of HepIm enabled the researchers to characterize various features of PXE patho-biochemistry.
In general, the manuscript is well structured, clear to the reader and transfers the state-of-the-art knowledge on PXE to the reader in adequate transparency.
Specific comments:
- The manuscript offers a broad selection of technologies, including CRISPR-Cas9 mediated hABCC6 knock-out, genomic DNA amplification, T7 endonuclease/mutagenesis assay, TA cloning for single allele sequencing, and sanger sequencing to introduce the plasmid and characterize the clones. Immunofluorescent staining, and other basic assays were used to further characterize the clones.
- The aim to develop an ABCC6-deficient hepatocyte cell culture system was fulfilled by using a CRISPR Cas9-based genome editing system.
- Images are given in excellent quality, a bit overloaded and hard to read but in general clear.
- Evidence of remodeling of ECM with the expression of COL1A1 and DCN was found. Markers to analyze the senescence status, including p21 and SA-b-Gal and the senescence-associated inflammatory status including IL1b, IL6 as well as C3 were presented and discussed, but without the end stage of fibrosis. This can be addressed more prominently with this evidence.
- Please include some limitation discussion.
I really like your work and I am not trying to discourage you and your efforts but please be a bit more visionary.
Round 2
Reviewer 1 Report
Comments and Suggestions for Authors
Dear authors,
Thank you for your response. However, I must express my disappointment that the key concerns regarding the documentation of gene-editing events and ABCC6 validation by Western blot have not been addressed. I understand the challenges regarding antibody availability but remain concerned about the low gene-editing efficiency and the lack of sufficient validation to confirm ABCC6 loss.
